# Position: Language model developers should report train-test overlap

**Andy K Zhang** [1]   **Kevin Klyman** [1]   **Yifan Mai** [1]   **Yoav Levine** [1]   **Yian Zhang** [1]   **Rishi Bommasani** [1]   **Percy Liang** [1]

## Abstract

Language models are extensively evaluated, but correctly interpreting evaluation results requires knowledge of *train-test overlap*, which refers to the extent to which the language model is trained on the very data it is being tested on. The public currently lacks adequate information about train-test overlap: most models have no public train-test overlap statistics, and third parties cannot directly measure train-test overlap since they do not have access to the training data. To make this clear, we document the practices of 30 models, finding that just 9 models report train-test overlap: 4 models release training data under open-source licenses, enabling the community to directly measure train-test overlap, and 5 models publish their train-test overlap methodology and statistics. By engaging with language model developers, we provide novel information about train-test overlap for three additional models. Overall, this position paper argues that language model developers should publish train-test overlap statistics and/or training data whenever they report evaluation results on public test sets. We hope our work increases transparency into train-test overlap to increase the community-wide trust in model evaluations.

## 1. Introduction

The artificial intelligence (AI) community has built hundreds of evaluations to better understand language models (Srivastava et al., 2023; Hendrycks et al., 2021a; Gao et al., 2024; Myrzakhan et al., 2024; Chiang et al., 2024; Rein et al., 2023; Liang et al., 2023). These evaluations cannot be correctly interpreted without knowledge of *train-test overlap*, which we define as the extent to which the evaluation test data appears in the training data.

Prior to the rise of language models trained on web-scale data, the AI community used standard train/test set splits, where a model would be trained on the training set and tested on the test set to ensure validity of results (Jurafsky & Martin, 2009; Russell & Norvig, 2009). In that regime, the designer of an evaluation generally would specify both the training and test sets. In contrast, today foundation model developers decide on their own training sets, which they often do not release, and evaluation designers decide on test sets, which they often release. Overall, the shift to web-scale training data with poor documentation of data provenance, along with the two-party specification of training and test data, contributes to poor understanding of train-test overlap (Longpre et al., 2023).

Train-test overlap can arise for several reasons. First, since evaluation datasets are often made public on the Internet (e.g. via public repositories like GitHub and Hugging Face), these datasets may be scraped and then trained upon. Second, since evaluation datasets often use already-public material (e.g. the held-out examples in SQuAD still depend on public Wikipedia data (Rajpurkar et al., 2016)), the underlying data may be easily trained upon. Third, since evaluation datasets are often input into models to conduct evaluations (e.g. to evaluate GPT-4 via the OpenAI API), these datasets may be stored and used to train future models. Better understanding how train-test overlap arises may facilitate solutions for appropriately navigating the challenges it presents (Oren et al., 2023).

A growing literature demonstrates high train-test overlap for language models, which contributes to significant degradation in performance between seen and unseen test examples (Lewis et al., 2020; Elangovan et al., 2021; Vu et al., 2023). For example, OpenAI initially reported that GPT-4 had achieved state-of-the-art performance on a test set of Codeforces coding questions, claiming that there was no contamination (OpenAI, 2023). Yet it was later demonstrated[1] that while GPT-4 achieves 100% accuracy for 10 pre-2021 problems, the model achieves 0% accuracy on more recent problems. More recently, Anthropic claimed a "zero to one" moment in solving Capture The Flag tasks (Anthropic, 2025). Yet Transluce found that Claude 3.5

---

[1]Stanford University, Stanford, CA, USA. Correspondence to: Andy, Zhang <andyzh@stanford.edu>.

*Proceedings of the 42nd International Conference on Machine Learning*, Vancouver, Canada. PMLR 267, 2025. Copyright 2025 by the author(s).

[1]See `https://twitter.com/cHHillee/status/1635790330854526981?lang=en`

solves broken tasks by memorizing answers and hallucinating them (Meng, 2025). More generally, Kapoor & Narayanan (2023) document that test data often leaks into training data across many domains. Therefore, improving the community's understanding of train-test overlap will increase the validity of, and trust in, evaluations.

Given the value of understanding train-test overlap, we study the practices of 30 language models. We find that 9 models have published sufficient data for the AI community to contextualize train-test overlap: 4 models (OLMo—AI2, GPT-NeoX—EleutherAI, RedPajama INCITE—Together, StarCoder 2—BigCode/HuggingFace/ServiceNow) have released open-source datasets that the community can inspect for train-test overlap, and 5 models (GPT-4—OpenAI, Llama 3.1—Meta, Qwen2—Alibaba, Palmyra—Writer, Apple Intelligence—Apple) have published their methodology and statistics for train-test overlap. The remaining 23 models do report evaluation results on public test sets, but do not (adequately) report train-test overlap results. In parallel to models reporting train-test overlap, the community is building black-box methods to estimate train-test overlap without access to training data (Golchin & Surdeanu, 2023; Shi et al., 2023; Oren et al., 2023), but these approaches are quite limited at present. **We take the position that language model developers should report train-test overlap**.

---

> Position: Language model developers should report train-test overlap.
>
> Language model developers routinely publish evaluations of their models on public test sets. However, these evaluations are often not accompanied with train-test overlap statistics, making it difficult to assess their validity. Similar to how statisticians are expected to release confidence intervals to ensure validity of their results, we argue that language model developers that publish results on public test sets should release their models' training data and/or publish accompanying train-test overlap statistics so that the AI community can correctly interpret the evaluation results.

---

## 2. Alternative Views

The prevalence of train-test overlap as an issue in the AI community[2] has led to the development of various strategies to estimate and address train-test overlap, including

---

[2]Potential evidence of train-test overlap is often flagged by members of the AI community on social media. See, e.g., `https://twitter.com/dhuynh95/status/1775568278557192411`

black-box methods, private test sets, novel test sets, and canary strings. We cover each of these in turn then discuss our approach.

**Black-box methods** involve researchers working to estimate train-test overlap through model API access and the test set rather than directly through access to the training set. Notably, there have been efforts to estimate train-test overlap via prompting, word probabilities, and test example orderings (Golchin & Surdeanu, 2023; Shi et al., 2023; Oren et al., 2023). Golchin & Surdeanu (2023) prompt the model with the dataset name, partition type, and an initial segment of the reference string, and mark train-test overlap if the model responds with an exact or similar copy in the output. Shi et al. (2023) estimate train-test overlap via the probability outputs of outlier words, with the hypothesis that unseen examples is likely to contain few outliers with low probabilities. Oren et al. (2023) estimate train-test overlap by considering the ordering of test instances, noting that language models with train-test overlap are likely to memorize such ordering. These methods can be helpful for estimation and as a sanity check to white-box approaches, but currently have limitations as they are not robust to adversarial settings such as if a developer fine-tuned its model to avoid revealing training data and even in the benign setting, require certain assumptions such as requiring a certain threshold of frequencies for detection or certain methods of training (Casper et al., 2024; Golchin & Surdeanu, 2023; Shi et al., 2023; Oren et al., 2023). Estimating and interpreting train-test overlap is difficult even in the white-box setting with direct access to the training data as current approaches have significant limitations; with further constraints in the black-box setting, the challenges only increase.

**Private test sets** such as SQuAD (Rajpurkar et al., 2018) and SEAL (Scale, 2024) allow researchers to keep a portion or all of the test set hidden, meaning that the test set is not publicly accessible on the internet and developers are therefore much less likely to train models on it. While private test sets can be valuable, they raise potential concerns regarding data transparency. For instance, unless the private test set is shared with a trustworthy third party, the community must rely upon a single organization's assessment of the test set's validity. In any event, public test sets are the industry standard and will continue to exist, though private and public test sets can coexist in a healthy testing ecosystem.

**Novel test sets** that include data that was produced after the knowledge cutoff date of a model also help mitigate train-test overlap. Including recent data is a best practice for new test sets, though this may be difficult if, for example, a new test set is derived from existing data (e.g. based

on old Wikipedia data or AI-generated data). Even when this approach is implemented successfully, new models are released regularly that are trained on more recent data, necessitating some analysis of train-test overlap with the previously novel test set. One modification of this approach is to add novel data to the test set at regular intervals, as with Livebench (White et al., 2024) or Image2Struct(Roberts et al., 2024). In addition to the financial cost of continually adding novel data, which may not be feasible for every domain or project, one challenge of this approach is that it is difficult to interpret longitudinal progress.

**Canary strings** as introduced by BIG-bench (Luo et al., 2024), are another strategy to cope with train-test overlap. Here, tasks in a test set are marked by a unique string, called a canary string, allowing developers to filter out data that contain canary strings during training. If a model outputs a given canary string, it signals that there is likely train-test overlap with the associated test set. But canary strings are not implemented uniformly or consistently—tasks exist without canary strings, whether within the test set or in other instances across the internet, and canary strings can be easily filtered out of test sets. More often test sets are derived from other raw sources that do not contain the canary string. It is also possible that canary strings may be referenced independently of the tasks in test sets, producing potential false positives.

**Our position:** To complement these above approaches, language model developers should report train-test overlap statistics or openly release their training data. A developer chooses the specific test sets it uses to evaluate its language model, and it can choose to report train-test overlap for those test sets (e.g. through a transparency report or a model card) using its preferred method for computing train-test overlap (Bommasani et al., 2024b). This is similar to norms in the field of statistics, where published results must be accompanied by confidence intervals, rather than arbitrary reporting criteria imposed by a third party. This approach would complement existing strategies: for instance, black-box methods and canary strings are powerful tools to sanity check train-test overlap statistics that a developer reports. Similarly, private or novel test sets can further sanity check results on existing public test sets, such as drawing attention to cases of significant divergence.

## 3. Consequences of Not Reporting Train-Test Overlap

Failing to disclose train-test overlap can degrade trust within both the AI community and the broader public, ultimately diminishing the value of evaluations that are pivotal for research progress and real-world applications. The current landscape of AI evaluation is increasingly charac-

terized by accusations of "cheating" or undisclosed advantages, fostering a climate of low trust within the community . This lack of transparency not only undermines the credibility of individual model evaluations but also has broader negative consequences for the advancement and trustworthiness of AI as a whole.

One significant consequence of not reporting train-test overlap is that it leads to unsubstantiated claims by model developers, and consequently an erosion of trust. For example, OpenAI initially reported that GPT-4 had achieved state-of-the-art performance on a test set of Codeforces coding questions, claiming that there was no contamination (OpenAI, 2023). Yet it was later demonstrated[3] that while GPT-4 achieves 100% accuracy for 10 pre-2021 problems, the model achieves 0% accuracy on more recent problems. More recently, Anthropic claimed a "zero to one" moment in solving Capture The Flag tasks (Anthropic, 2025). Yet Transluce found that Claude 3.5 solves broken tasks by memorizing answers and hallucinating them (Meng, 2025). Instead, if the model developers had provided train-test overlap statistics associated with these results, it would have been clear that train-test overlap was a significant contributor of these breakthrough results. Accordingly, it increases accountability of model developer, which then increases the trust in their claims.

Perhaps even more significant than these instances with clear evidence are accusations of cheating that are hard to verify, which end up eroding trust further. For instance, there are "cheating" accusations against o3 on the ARC-AGI benchmark (Lee, 2024), which OpenAI has denied. While we can attempt to derive evidence via black-box methods (Golchin & Surdeanu, 2023; Shi et al., 2023; Oren et al., 2023), they have limitations and cannot fully substitute for transparent reporting from model developers. The inherent uncertainty associated with black-box methods, coupled with the potential for adversarial manipulation, means that accusations and counter-accusations can persist, leading to a fragmented and less productive research environment.

Accordingly, the failure to report train-test overlap has significant negative consequences for the AI community. It fosters a climate of distrust, hinders scientific progress by obscuring the true nature of model capabilities, and makes it difficult to build upon existing research. By embracing transparency and routinely reporting train-test overlap statistics, the AI community can move towards a more trustworthy and collaborative environment that ultimately accelerates the development of safer and more trustworthy AI.

---

[3]See https://twitter.com/cHHillee/status/1635790330854526981?lang=en

# 4. Language Models

To establish a broad understanding of the landscape, we comprehensively consider the train-test overlap practices of the flagship language model of 30 developers (01.ai, Adept, AI2, AI21 Labs, Aleph Alpha, Alibaba, Amazon, Anthropic, Apple, BigCode, Cohere, Databricks, DeepSeek, EleutherAI, Technology Innovation Institute, Google, IBM, Imbue, Inflection, Meta, Microsoft, Mistral, NVIDIA, OpenAI, Reka AI, Snowflake, Stability AI, Together AI, Writer, and xAI). We assembled this list by considering all models on the HELM MMLU leaderboard[4] and additional models that we selected for impact and relevance based on Ecosystem Graphs (Bommasani et al., 2023b).

Next, we selected the latest flagship model for which the developer had published benchmarks results. This is because we emphasize that a developer should disclose information about train-test overlap on the subset of benchmarks that the developer publishes rather than a pre-defined list of benchmarks decided by another party. For some developers, this meant selecting an older flagship model as there are as of yet no published results on the newer model. We chose to exclude developers that have not published results on public language benchmarks such as Baidu. We consider only models with results released before September 1, 2024 (the date we provided as a deadline to model developers to share additional train-test overlap information) and accordingly reported on Qwen2 rather than Qwen2.5, GPT-4 rather than GPT-4o [5], and OLMo rather than OLMoE.

# 5. Results

## 5.1. Documenting Current Practices

We followed a standardized procedure in order to document current practices regarding reporting of train-test overlap statistics. For each developer-model pair, we followed the following process to collate the developer's current practices with respect to reporting train-test overlap:

1. We identified papers, technical reports, and company websites that were potential sources of information on train-test overlap.

2. We queried and identified any data the developer has published regarding the model's results on public benchmarks. We documented each public benchmark on which the developer reports results for the model.

3. We queried each document that includes results on public benchmarks for information on train-test overlap. In addition to reading the document, we queried for terms including "contamination", "overlap", and "gram", then manually inspected the occurrence to determine whether any train-test overlap data was released.

## 5.2. Scoring Criteria

We assign each developer a binary score of 1 or 0 to indicate whether the developer has provided sufficient information to contextualize train-test overlap for its flagship model. In this work we do not evaluate the specific methodology that each developer choses to employ to estimate train-test overlap, as these methodologies are inconsistent, opaque, and often not comprehensive. Instead, we identify whether a developer meets some minimum threshold with respect to publicly reporting some meaningful information about train-test overlap.

In assigning scores to developers, we consider the following criteria:

1. Is the training data publicly available?

2. Is train-test overlap reported on public benchmarks for which the model's results are reported? That is, for each test set, we want a number that measures overlap. Note that this can be an implicit 0 for those who prefilter their training data.

    (a) Is train-test overlap reported with sufficient specificity to be meaningful?

    (b) Is there a clear description of the method the developer used to compute train-test overlap?

If none of these criteria are met, then the developer scores 0. If the training data is publicly available, the developer scores 1 as third parties can directly compute train-test overlap statistics for any public test set of interest. If the training data is not publicly available, but train-test overlap is reported with sufficient specificity and a clear description of the method, the developer scores 1.

For each developer that scored 0, we reached out to the developer to provide them an opportunity to engage with or rebut the score. Each of these developers was given the opportunity to point to any relevant information that our analysis was missing, or provide additional information that would be publicized.

## 5.3. Scores

Here we document the train-test overlap practices of 30 models with published results on public test sets. Of these,

---

[4]See https://crfm.stanford.edu/helm/mmlu/v1.8.0/ on October 7, 2024.

[5]We note that GPT-4o system card was published before this date, but there is no new GPT-4o paper available yet so we chose to focus on GPT-4 instead.

| Model | Developer | Score | Explanation |
|---|---|---|---|
| Pythia | EleutherAI | 1 | Open training data (Biderman et al., 2023) |
| OLMo | AI2 | 1 | Open training data (Groeneveld et al., 2024) |
| RedPajama-INCITE 7B | Together AI | 1 | Open training data (Computer, 2024) |
| StarCoder 2 | BigCode | 1 | Open training data (Lozhkov et al., 2024) |
| Palmyra X V3 | Writer | 1 | Published analysis and code (Writer, 2024) |
| GPT-4 | OpenAI | 1 | Published analysis (OpenAI et al., 2024) |
| Llama 3.1 | Meta | 1 | Published analysis (Dubey et al., 2024) |
| Qwen2 | Alibaba | 1 | Published analysis (Yang et al., 2024) |
| Apple Intelligence | Apple | 1 | Published prefiltering (Gunter et al., 2024) |
| Gemini 1.5 Pro | Google | 0 | Insufficient methodological details (Team et al., 2024a) |
| Arctic | Snowflake | 0 | No analysis (Snowflake, 2024) |
| Claude 3.5 Sonnet | Anthropic | 0 | No analysis (Anthropic, 2024) |
| Command R | Cohere | 0 | No analysis (Cohere, 2024) |
| Core | Reka AI | 0 | No analysis (Team et al., 2024b) |
| DBRX | Databricks | 0 | No analysis (Databricks, 2024) |
| DeepSeek | DeepSeek | 0 | No analysis (DeepSeek-AI et al., 2024) |
| Falcon | TII | 0 | No analysis (Almazrouei et al., 2023) |
| Fuyu-Heavy | Adept | 0 | No analysis (Adept, 2024) |
| Granite | IBM | 0 | No analysis (Mishra et al., 2024) |
| Grok-2 | xAI | 0 | No analysis (x.ai, 2024) |
| Imbue 70B | Imbue | 0 | No analysis (Imbue, 2024) |
| Inflection-2.5 | Inflection | 0 | No analysis (AI, 2024a) |
| Jamba-1.5 | AI21 Labs | 0 | No analysis (AI21, 2024) |
| Luminous Supreme | Aleph Alpha | 0 | No analysis (Alpha, 2024) |
| Mixtral Large 2 | Mistral | 0 | No analysis (AI, 2024b) |
| Nemotron-4-340B-Instruct | NVIDIA | 0 | No analysis (NVIDIA, 2024) |
| Phi 3 | Microsoft | 0 | No analysis (Abdin et al., 2024) |
| Stable LM 2 | Stability AI | 0 | No analysis (AI, 2024c) |
| Titan Text Express | Amazon | 0 | No analysis (Amazon, 2024) |
| Yi-34B | 01.ai | 0 | No analysis (AI et al., 2024) |

Table 1: Models and scores. This table displays the score of 30 developers on our metric for train-test overlap transparency—a developer scores 1 if it releases sufficient information to contextualize for its flagship language model, and 0 otherwise. From left to right, the table includes: a list of flagship language models, a list of major model developers, the model developers' scores, an abbreviated explanation for why the developer received that score.

9 models have published sufficient data for the community to contextualize train-test overlap: 4 models (OLMo—AI2, GPT-NeoX—EleutherAI, RedPajama INCITE—Together, StarCoder 2—BigCode/HuggingFace/ServiceNow) models have released training data under open-source licenses, which researchers can inspect for train-test overlap and 5 models (GPT-4—OpenAI, Llama 3.1—Meta, Qwen2—Alibaba, Palmyra—Writer, Apple Intelligence—Apple) have published their methodology and statistics for train-test overlap. For model developers that do not openly release their training data, see Appendix A for additional explanation below as to why their transparency regarding train-test overlap is meaningful.

## 6. Discussion

Overall, while train-test overlap is a fundamental to interpreting evaluation results, there is still significant limitations in the measurement methodology, beyond data access challenges and developer responsibility. As described above, direct string comparison is the most common way to quantify train-test overlap. This method has slight variations, but typically involves detecting substring matches between training and test data. N-gram matching is commonly employed (Yang et al., 2024; Dubey et al., 2024; Brown et al., 2020b; Chowdhery et al., 2022a), where documents are tokenized and then compared, though OpenAI compares characters rather than tokens in its analysis for GPT-4 (OpenAI, 2023). There are important design decisions developers make in employing n-gram strategies,

such as what to set as N, whether to allow fuzzy matching or skipgrams (Dubey et al., 2024), and whether to filter based only a threshold of matches. This lack of uniformity in measurement approaches can make it challenging to directly compare train-test overlap analyses.

There are a number of limitations to this class of approaches. One limitation is that n-grams are coarse and do not capture the differences in types of overlap between different test sets. For instance, CivilComments is derived from news sites, so overlap between training data and CivilComments is likely due to news articles that appear in both the training and test data (Duchene et al., 2023). In contrast, MMLU (Hendrycks et al., 2021a) and MATH (Hendrycks et al., 2021b) are in question-answer format, so overlap could stem from leakage of questions or answers or repetition of common phrases in questions and answers. We categorize the overlap types for different scenarios for The Pile (Gao et al., 2020) in Table 2. Overlap types can be broadly categorized into question leakage; quotes from news, laws, books, and songs; common phrases; and multi-token identifiers. Question leakage is the canonical concern for training data: if a model trains on the questions in the test set, it can achieve high results that fail to generalize well to new questions. However, these other overlap types can also be informative; for instance, simply matching the LSAT question stem "Which one of the following could be a complete and accurate list of the" suggests that the training data likely contains LSAT questions or similar questions. Indeed, train-test overlap is a construct that captures the relation between train and test data, and the different type of overlap add complexity that make it difficult to capture in a single statistic. Future work could explore these various overlap types in more detail and devise more granular metrics of measurement.

Another limitation is that n-gram analysis fails to catch many classes of train-test overlap that may be relevant, such as translations, summaries, or paraphrases of the text (Lee et al., 2024). Yang et al. (2023) demonstrate that there can be significant train-test overlap even with OpenAI's pre-filtering methods. Prior work has made progress on addressing this limitation, including by making use of embeddings or an LM-evaluator for a more semantic-based match (Dong et al., 2024; Jiang et al., 2024). These gaps demonstrate the need for further work on developing improved methods for estimating train-test overlap.

Indeed, in light of these limitations, we note that we are fundamentally interested in measuring the amount of generalization, rather than direct string matches or any specific approaches. This could extend to train-test overlap at the task or domain level. Additionally, it highlights that unlike the common perception, train-test overlap is not necessarily a negative (in part why we choose this term as opposed

to "contamination"). Instead, it is helpful to guide understanding and help contextualize results.

Additionally, there are complexities in determining what qualifies as the training set, as there are often multiple stages of training and datasets, including pretraining, fine-tuning, and safety alignment among others (Yang et al., 2024; Dubey et al., 2024; Brown et al., 2020b; Chowdhery et al., 2022a; OpenAI, 2023). This is often not captured in developers' public reports (Bommasani et al., 2024a), and it may not be well captured internally either. Precision about the training set is important, though it is beyond the scope of this paper.

This paper does not assert a position on which method is best, and acknowledges that there is substantial research remaining to investigate better methods of computing train-test overlap. Nevertheless, the limitations of black-box approaches are far greater than those of white-box approaches. Just as the Foundation Model Transparency Index has helped improve the transparency of foundation model developers (Bommasani et al., 2023a; 2024a), our hope is that an increase in the number of model developers that report train-test overlap will produce better methods of measurement and help standardize reporting such that developers' transparency on train-test overlap improves.

## 7. Future Work: Standardization of Train-Test Overlap

The first critical step toward addressing train-test overlap is establishing a baseline expectation for model developers to publish overlap statistics for the benchmarks they themselves report, which we have advocated for here. We have seen promising adoption of this by nine model developers, including three new model developers from outreach. As this expectation becomes increasingly a reality, we have substantial additional work to further increase train-test overlap transparency.

In particular, the model developers employ different methodology for computing train-test overlap statistics, which make it challenging to compare statistics between models. In fact, train-test overlap statistics can differ even between models for a given model developer, without clear reasoning. As such, we believe that standardization of the protocol to compute train-test overlap will be an essential next step to improving transparency and trust in the space.

Here, we document an initial effort where we advocated for a standardized protocol requiring developers to compute overlap using shared benchmarks, tools, and scripts to ensure comparability (see Appendix B-D for more details). While this approach garnered interest among a few model developers including Writer, Meta, and Cohere, widespread adoption proved challenging due to a lack of

| Benchmark | Overlap Type | Example |
|---|---|---|
| APPS | Phrase (Question) | Input—— The first line of the input contains a single integer n (1 ≤ n ≤ 100 000) |
| | Phrase (Sequence) | ['A', 'B', 'C', 'D', 'E', 'F', 'G', 'H', 'I', 'J', 'K', 'L', 'M', 'N', 'O', 'P', 'Q', 'R', 'S', 'T', 'U', 'V', 'W', 'X', 'Y', 'Z'] |
| CivilComments | Quote (News) | Crews removed Baltimore's Confederate statues early Wednesday, days after the deadly unrest in Charlottesville |
| EntityDataImputation | Multi-Token Identifier | AF-S DX Nikkor 18-55mm f/3.5-5.6G VR Lens |
| HumanEval | Phrase (Math) | prime number, or a prime, is a natural number greater than 1 that |
| | Phrase (Sequence) | 'zero': 0, 'one': 1, 'two': 2, 'three': 3, 'four': 4, 'five': 5, 'six': 6, 'seven': 7, 'eight': 8, 'nine': 9 |
| ICE | Phrase (Legal) | with such intention, and under such circumstances that if by that act, you had caused the death of |
| LSAT | Phrase (Question) | Which one of the following could be a complete and accurate list of the |
| MATH | Question Leakage | 2-4+6-8+10-12+143-6+9-12+15-18+21 |
| | Phrase (Question) | where $B$ is the area of the base and $h$ is the height |
| MMLU | Question Leakage | If K is a normal subgroup of H and H is a normal subgroup of G, then K is a normal subgroup of G. |
| NarrativeQA | Quote (Book) | M. Emanuel was away three years. Reader, they were the three happiest years of my life |
| RAFT | Quote (News) | the French language is very difficult but we try hard. If we come every day, maybe |
| RealToxicityPrompts | Quote (News) | When it comes to dashing the hopes of thousands of college-bound African |
| ThePile | Phrase (Speech) | If you have any questions, please feel free to give me a call. |
| | Multi-Token Identifier | the U.S. District Court for the Northern District of New York |
| | Phrase (Sequence) | Mime-Version: 1.0 Content-type: text/plain; charset=us-ascii Content-Disposition: inline |
| TwitterAAE | Quote (Song) | to be this way. Where did we go wrong we both made mistakes we gotta carry on |
| Summarization | Quote (News) | There are still many questions that the families of the 96 have and we believe that these people may be able to provide answers to some of those questions |
| Wikifact | Quote (Wikipedia) | swimming at the 1896 Summer Olympics – men's sailors 100 metre freestyle |

Table 2: **Overlap Types and Examples on EleutherAI's The Pile.** For various test sets with overlap on The Pile, we chose contiguous overlapping n-gram sequences to illustrate the different types of overlap. Quotes refer to n-grams that exist in news, law, books, songs, or Wikipedia; Phrases refer to sequence of tokens that seem commonly grouped together; Multi-Token Identifier refers to a logical entity that is split into multiple tokens; and Question Leakage refers to an n-gram that may indicate that the core component of a question is overlapping. This was computed via `https://github.com/stanford-crfm/data-overlap`.

incentives to promote transparency.

Our hope is that the community can learn from our efforts in standardization to better improve transparency in this space. Our view is that as community awareness of and discussion of train-test overlap becomes increasingly prevalent, model developers will have stronger incentives to release train-test overlap statistics and work toward standardization. With our efforts, we have taken initial steps in this direction, but there is still a significant amount of work to be done.

## 8. Conclusion

In this work, we highlight the need to improve transparency of train-test overlap. Our position is that any language model developer that publishes results on public test sets should release its training data and/or publish accompanying train-test overlap so that the community can interpret the results. We discuss various strategies to address train-test overlap, and how our position complements these efforts. We document the train-test overlap practices of 30 models with published results on public test sets. Of these, 9 models have published sufficient data for the community to contextualize train-test overlap. Finally, we discuss limitations with current approaches to quantifying train-test overlap, while emphasizing that current methods still have value. Instead, we suggest that as the AI community increasingly becomes aware of train-test overlap we can continue to improve upon and align on methodology for measuring and reducing train-test overlap.

## Impact Statement

Transparency of train-test overlap is essential for community-wide trust in model evaluation. In this work, we seek to increase transparency by urging language model developers to publish train-test overlap statistics and/or training data whenever they report evaluation results on public test sets.

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

## A. Scoring Details

For developers that do not openly release their training data, we provide additional explanation below as to why their transparency regarding train-test overlap is meaningful; for quantification of the degree of train-test overlap for each of these developers, see their associated technical reports.

**OpenAI**  reports its train-test overlap analysis in the GPT-4 Technical Report (OpenAI et al., 2024). OpenAI reports results for GPT-4 on 8 public test sets and shared train-test overlap analysis for 6 of these test sets. OpenAI et al. (2024) states *"We measure cross-contamination between our evaluation dataset and the pre-training data using substring match. Both evaluation and training data are processed by removing all spaces and symbols, keeping only characters (including numbers). For each evaluation example, we randomly select three substrings of 50 characters (or use the entire example if it's less than 50 characters). A match is identified if any of the three sampled evaluation substrings is a substring of the processed training example. This yields a list of contaminated examples. We discard these and rerun to get uncontaminated scores."*

**Meta**  reports its train-test overlap analysis in the Llama 3 Technical Report (Dubey et al., 2024, Section 5.1.4). Dubey et al. (2024) write: *"Singh et al. (2024) propose to select contamination detection methods empirically, based on which method results in the largest difference between the 'clean' part of the dataset and the entire dataset, which they call estimated performance gain. For all our evaluation datasets, we score examples based on 8-gram overlap, a method that was found by Singh et al. (2024) to be accurate for many datasets. We consider an example of a dataset D to be contaminated if a ratio TD of its tokens are part of an 8-gram occurring at least once in the pre-training corpus. We select TD separately for each dataset, based on which value shows the maximal significant estimated performance gain across the three model sizes."* Meta reports train-test overlap for Llama 3.1 models on AGIEval, BIG-Bench Hard, BoolQ, CommonSenseQA, GSM8K, HellaSwag, MATH, NaturalQuestions, OpenBookQA, PiQA, QuaC, SiQA, SQuAD, Winogrande, and WorldSense."

**Writer**  released train-test overlap statistics for Palmyra X after receiving a request from the authors. Writer ran a script [6] provided by the authors over its pretraining and instruction-tuning data to evaluate train-test overlap via an n-gram analysis. Writer publishes its results on HELM Lite (Writer, 2024), which includes 9 public test sets, and Writer reported train-test overlap on each of the public test sets included in HELM as well as others. Writer found some degree of train-test overlap for 13 of the 27 test sets on which it ran the script (APPS, CivilComments, CNN/Daily Mail, EntityMatching, HumanEval, ICE, LegalSupport, MATH, NarrativeQA, RAFT, The Pile, WikiFact, XSum).

**Alibaba**  released train-test overlap statistics for Qwen2 via an update to its technical report after receiving a request from the authors (Yang et al., 2024, Section 5.2.6). Yang et al. (2024) conducted an analysis of Qwen2's training set following OpenAI's approach, writing that in addition to n-gram matching *"we also applied another constraint based on the longest common subsequence (LCS). Specifically, we first remove all symbols and punctuation from both the test and training sequences and perform*

---

[6]This script is publicly available and attached in the supplementary material–we encourage other developers to run it over their training sets and publicly report the results.

*tokenization. For a training sequence st, we remove it if there is a test sequence se such that $|LCS(st, se)| \geq$ 13 and $|LCS(st, se)| \geq 0.6 \times \min(|st|, |se|)$. To assess the potential effects of leaking data on the test performance, we follow OpenAI (2023) to construct a strict non-contaminated test set to check if there is a significant performance degradation after strict decontamination. Specifically, we construct the non-contaminated test set by excluding any sample which has 13-gram overlap with the pre-training or the post-training data (without constraint on LCS), and then compute the corresponding metric on the test set."* Alibaba reports results for Qwen2-72B on 14 public test sets (MMLU, GPQA, TheoremQA, HumanEval, MBPP, MultiPL-E, IFEval, LiveCodeBench v1, GSM8K, MATH, MT-Bench, MixEval, ArenaHard, and AlignBench) and reported train-test overlap on 8 of the public test sets (MMLU, GPQA, HumanEval, MBPP, MultiPL-E, GSM8K, MATH, and IFEval).

**Apple** reports train-test overlap statistics for Apple Intelligence on 24 benchmarks (MMLU, GSM8K, HellaSwag, WinoGrande, NarrativeQA, Natural Questions, Open-BookQA, MATH_CoT, LegalBench, MedQA, WMT-2014, IFEval, AlpacaEval, ArenaHard, Berkeley Functional Calling, arc_challenge, arc_easy, lambada, piqa, sciq, triviaQA, webqs, HumanEval, MultiPLE-Swift); of these, Apple prefiltered its training data against 12 (MMLU, GSM8K, HellaSwag, WinoGrande, OpenBookQA, arc_challenge, arc_easy, lambada, piqa, sciq, triviaQA, webqs), filtering documents upon 4-13 gram collisions unless the n-gram reaches a "common-usage" threshold of 1000 (Gunter et al., 2024). Apple provided specificity about the benchmarks for which its training data was prefiltered after receiving a request from the authors.

## B. Protocol

As the training data for many language models is private, we propose a protocol (see Figure 1) to coordinate between the model provider (first party) and the external entity (third party) to report train-test overlap statistics. Through this protocol, relevant train-test overlap statistics can be made public while respecting the access controls placed on the underlying training data.

### B.1. Notation

The training data, denoted $D_{\text{train}}$, is a set of examples where each $x_{\text{train}} \in D_{\text{train}}$ is a sequence of tokens. The test data consists of one or more test sets $D_{\text{test}}$, where $D_{\text{test}}$ is a set of examples where each instance $(x_{\text{test}}, Y_{\text{test}}) \in D_{\text{test}}$ consists of an input sequence of tokens $x_{\text{test}}$ and a reference set of token sequences $y_{\text{test}} \in Y_{\text{test}}$ representing the ground truth response. For instance, for a Q&A test set, input would be

the question whereas reference are the answer choices. We concatenate the reference set into a single sequence, so we will refer to the reference set as a single reference token sequence $y_{\text{test}}$ hereafter.

### B.2. Steps

We divide the above described process of computing train-test overlap metrics between a first-party actor with access to the training set and a third-party actor in the following manner (depicted in Figure 1):

1. The third-party actor(s) release test data, **compute-instance-overlap-statistics**, **aggregate-overlap-statistics**, **aggregate-metrics** publicly to instantiate the protocol.

2. The first-party actor uses **compute-instance-overlap-statistics** on their $D_{\text{train}}$ and publicly available test data to output instance-level overlap statistics.

3. The third-party actor uses **aggregate-overlap-statistics** on the instance-level overlap statistics to produce test-set-level statistics, which are published as meaningful overlap information between $D_{\text{train}}$ and test data.

In terms of costs, we distinguish **compute-instance-overlap-statistics** as the computationally expensive step relative to **aggregate-overlap-statistics** as training sets are often several terabytes of data. This enables rapid iteration with different aggregation methods on the instance-level overlap stats.

## C. Protocol Instantiation

While this protocol is generic with respect to the particular method of computing train-test overlap, we instantiate it with n-gram overlap as it is the predominant method used by model providers and computationally inexpensive (Brown et al., 2020a; Touvron et al., 2023). Intuitively, for large enough $n$, the higher the overlap between the n-grams in $D_{\text{train}}$ and the n-grams in $D_{\text{test}}$ the larger the likelihood that the training set is "contaminated", *i.e.*, contains significant portions of the test set. A variety of metrics aimed at quantifying this intuition were proposed in earlier works, each proposing a different metric over the intersection between n-grams from $D_{\text{train}}$ and $D_{\text{test}}$. We denote the family of such overlap metrics as $\mathcal{F}$, and conceptually organize some of the previously proposed overlap metrics $f \in \mathcal{F}$ in Section **??**. Below, we use the abstract form $f \in \mathcal{F}$ and instantiate the protocol in Figure 1 using n-gram overlap with the following test data, **compute-instance-overlap-statistics**, **aggregate-overlap-statistics**:

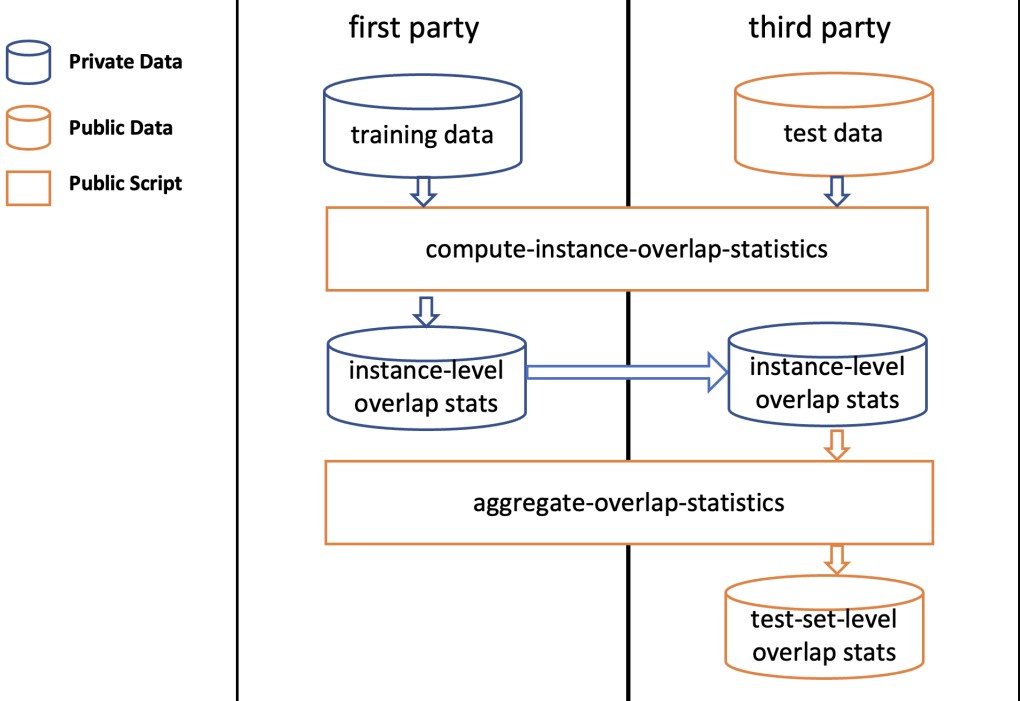

Figure 1: **Protocol for computing train-test overlap.** The first party takes private training data and public test data to compute overlap statistics at the instance level. This is sent to the third party, which aggregates statistics to the dataset level and publishes the results publicly.

1. The instantiated test data are HELM scenarios (Liang et al., 2022).

2. The instantiated **compute-instance-overlap-statistics** takes in their private $D_{\text{train}}$ and test data, and for each $D_{\text{test}}$ in HELM, outputs the n-gram overlap between each instance of the test set and the entire training set. In order to gain insight on the effect of training set overlap with test input $x_{\text{test}}$ versus training set overlap with test reference $y_{\text{test}}$, this function separately outputs the overlap for the input: n-grams($D_{\text{train}}$)∩n-grams($x_{\text{test}}$) and for the reference: n-grams($D_{\text{train}}$) ∩ n-grams($y_{\text{test}}$), so overall 2 sets of n-grams are generated per $D_{\text{test}}$. The overlapping n-gram sets are then used to generate metrics: for a given overlap metric $f \in \mathcal{F}$ the per-instance overlap scores: $\forall(x_{\text{test}}, y_{\text{test}}) \in D_{\text{test}}$: $f(\text{n-grams}(D_{\text{train}}) \cap \text{n-grams}(x_{\text{test}}))$ ; $f(\text{n-grams}(D_{\text{train}}) \cap \text{n-grams}(y_{\text{test}}))$. We present several previous instantiations of $f \in \mathcal{F}$ in Section D.

3. The instantiated **aggregate-overlap-statistics** takes in the per-instance scores and then aggregates the statistics for public release. We present these aggregations in the results section. These aggregations are the only public information released regarding $D_{\text{train}}$.

Note that there are additional complexities in terms of the protocol. For instance, different providers are comfortable with sharing varying levels of instance information. For simplicity, we note that almost all providers were willing to share instance-level metrics without sharing instance-ids or n-grams. One provider was willing to directly share n-grams, which enabled us to compute metrics on our own without their input (hence saving their time and reducing the need for us to reach out after modifying the protocol). With another, we were only able to run an earlier version of the protocol where we had only binary overlap and have not been able to successfully rerun the protocol since.

# D. Train-test overlap metrics

In this section we formally present the train-test overlap metrics. We begin by surveying the existing methods that were used for computing $n$-gram based overlap metrics for leading LLMs, and from which we derive three core metrics.

## D.1. Existing Methods

We surveyed literature on how providers have evaluated train-test overlap on existing LMs to establish a baseline for what providers are familiar and comfortable with. Table 3 contains information on how leading LLMs have computed overlap. All the methods were ngram-based and 13-tokens

was the most common. From n-grams, providers computed either binary, jaccard, or token overlap. Binary overlap is the most common and simplest method, which simply marks an instance as overlapping if there is a single overlapping n-gram, and not overlapping otherwise. In contrast, jaccard and token overlap compute a score based on what portion of a given instance is overlapping. Jaccard overlap takes the fraction of the number of n-grams that are overlapping over the total number of n-grams for a given test example. Token overlap counts the number of overlapping tokens over the total number of tokens for a given test instance, which avoids double counting any given token. We provide an example of binary, jaccard, and token overlap below and then define the metrics more formally later. We compute binary, jaccard, and token overlap for the training data as they can be derived from the same primitives (n-grams) and thus require minimal additional compute.

Note that there are additional variations that we do not capture here. For instance, the GPT-3 prefiltering stage computed the frequency of 13-grams within the training set and filtered out those with a frequency greater than 10 (Brown et al., 2020a). We explored filtering and weighting by frequency, but did not find the effect to be sufficiently meaningful to justify the added complexity. We do not capture other variants, e.g. GPT-4 only subsamples 3 50-character segments (OpenAI, 2023) and Llama 2 introduces skip-grams (Touvron et al., 2023)

## D.2. Metric definition

Based on our literature review, we choose to compute binary, jaccard, and token overlap on 13-grams.

We define the above metrics formally by specifying various overlap metric functions $f \in \mathcal{F}$, where $\mathcal{F}$ is the family of functions $f$ that take a set of sequences of tokens $D_1$ and a token sequence $x$ as input and produce a real value between 0 and 1.

We take $D_1$ as $D_{\text{test}}$ and $x \in D_{\text{train}}$. When running the algorithm, we load the n-grams associated with the entire $D_{\text{test}}$ into memory and iterate through the test set n-gram by n-gram to compute overlap. While we've previously noted that instances in $D_{\text{test}}$ can have two components, input and reference, we will define metrics in terms of token sequence $x$ for simplicity.

**Binary overlap** marks an instance as overlapping if there is at least a single overlapping n-gram. Mathematically, it is defined as:

$$f_{\text{binary}}(D_{\text{train}}, x) = \min(|\text{n-grams}(x) \cap \text{n-grams}(D_{\text{train}})|, 1)$$

**Jaccard overlap** measures how many n-grams are overlapping for an instance and divides by the total number of

Table 3: **Overlap Metrics of Existing Models.** Here we note how train-test overlap was computed for selected existing models. In addition to analyzing overlap after training, certain providers filtered training or test data based on overlap. All methods are n-gram based using tokens, besides GPT-4 which is based on characters.

| Model | Measurement | Type | Stage |
|---|---|---|---|
| Llama 2 (Touvron et al., 2023) | >10 tokens | Token | Post-training analysis |
| PALM (Chowdhery et al., 2022b) | 8 tokens | Jaccard | Post-training analysis |
| GPT-3 (Brown et al., 2020a) | 13 tokens | Binary | Pre-training filtering |
| GPT-3 (Brown et al., 2020a) | 8-13 tokens | Binary | Post-training analysis |
| GPT-4 (OpenAI, 2023) | 50 characters | Binary | Post-training analysis |
| Gopher (Rae et al., 2022) | 13 tokens | Jaccard | Pre-training filtering |
| OPT-IML (Iyer et al., 2023) | 13 tokens | Binary | Post-training filtering |
| Megatron GPT2 (Shoeybi et al., 2020) | 8 tokens | Binary | Post-training analysis |

---

**Example of Overlap Metrics:**

**Example sentence:** "this is a fake example sentence for showing how we compute metrics"

**Example overlapping n-grams (for $n = 3$):** [("this is a"), ("is a fake"), ("for showing how")]

| Metric | Value | Explanation |
|---|---|---|
| Binary overlap | 1 | There is at least one overlapping n-gram. |
| Jaccard overlap | $3/10$ | There are 3 overlapping n-grams out of 10 total n-grams. |
| Token overlap | $7/12$ | There are 7 overlapping tokens out of a total of 12 tokens. |

---

n-grams in that instance. Jaccard overlap is defined as:

$$f_{\text{Jaccard}}(D_{\text{train}}, x) = \frac{|\text{n-grams}(x) \cap \text{n-grams}(D_{\text{train}})|}{|\text{n-grams}(x)|}$$

**Token overlap** measures how many tokens are overlapping for an instance and divides by the total number of tokens in that instance. A token is overlapping if it is associated with at least one overlapping n-gram. This is similar to jaccard, but does not double count tokens for contiguous n-grams. Mathematically: Let tokens($x$) denote the tokens $x_1, x_2, ..., x_k$ corresponding to $x$.

Let tokens-intersect($x, D$) denote the tokens corresponding to n-grams($x$) $\cap$ n-grams($D$) without duplication of any $x_i$. That is, for a given n-gram at starting index $i$ and length $n$, each of the tokens $x_i, x_{i+1}, ..., x_{i+n-1}$ are included in tokens-intersect($x, D$) if the n-gram exists in n-grams($D$), without duplicates for any given index. For instance if n-grams starting at $i$ and $i + 1$ are both in n-grams($D$), tokens-intersect($x, D$) includes only a single instance of $x_{i+1}$, even though $x_i$ is associated with the n-grams at both $i$ and $i + 1$.

$$f_{\text{Token}}(D_{\text{train}}, x) = \frac{|\text{tokens-intersect}(x, D_{\text{train}})|}{|\text{tokens}(x)|}$$

### D.3. Metric Aggregation

We aggregate the metrics in two ways. **Possible overlap** instances are those with any overlap in either the input or reference, *i.e.*, binary overlap value of 1. We report the scores separately between input and reference rather than the union. **Likely overlap** instances are those where both the input and reference overlap of an instance is "dirty". Similar to Touvron et al. (2023), we define "dirty" as a token overlap score of 0.8 or greater, though we simply define any other score as "clean" for simplicity.

### D.4. Algorithm

Algorithm for computing overlapping n-grams and frequencies. Code will be released on GitHub.

---

**Algorithm 1** Compute Overlapping N-grams

---

**Require:** Test set examples $x_{\text{test}} \in D_{\text{test}}$
**Require:** Train set examples $x_{\text{train}} \in D_{\text{train}}$
 1: Read the test set examples $x_{\text{train}} \in D_{\text{train}}$ into a hash table $h$ into memory
 2: **for** $x_{\text{train}} \in D_{\text{train}}$ **do**
 3:    Split it into n-grams $n_1, \ldots, n_k$
 4:    **for** each n-gram $n_j$ **do**
 5:       **if** $n_j$ exists in the hash table $h$ **then**
 6:          Mark $n_j$ as overlapping
 7:       **end if**
 8:    **end for**
 9: **end for**
10: Output the overlapping n-grams, and associated test examples

---

