# OpenReview forum: "Position: Language model developers should report train-test overlap"
_ICML.cc/2025/Position_Paper_Track — ICML 2025 Position Paper Track spotlightposter_

### Official Review · Reviewer_2Bmm · 2025-02-13

**Significance:** 4
**Argument Clarity:** 3
**Rating:** 5
**Confidence:** 4

**Questions:**

What would be useful next steps to operationalize train-test development? What criteria should be reported?

**Discussion Potential:**

3

**Paper Summary:**

This paper examines the problem of train-test contamination, where test data is inadvertently included in the training set, arguing that reported performance metrics for large language models can be misleading without transparency about such overlap. The authors survey 30 models to see whether developers either (1) release open training data allowing the community to check for contamination themselves, or (2) publish analyses of train-test overlap, finding that overall only 9 models provide information. They therefore urge developers to either share training data or at least publish robust contamination metrics for any public benchmarks on which they report results, ensuring that the broader community can interpret model evaluations.

**Position:**

Yes

**Position In Title:**

Yes

**Related Work:**

3

**Strengths And Weaknesses:**

Strenghts:
- Detailed Discussion of Alternative Approaches and how train-test-overlap could fit in: The paper acknowledges various methods for mitigating contamination (black-box detection, private/novel test sets, and canary strings) yet underscores that these approaches complement rather than replace developer-provided overlap information, and provide a good disucssion for this.
- Relevance: Given that a lot of governance, investment and product decisions are based on evaluations and benchmarks, this is a critical topic for discussion. Seems like a good step and best practice for the ML community to adopt
- Methods: They clearly explained what models were chosen and why, how the analysed the documents and how they derived an analysis
- Discussion of challenges: The authors discuss some of the challenges that come with both the release of the data an

Weaknesses:
- Next steps: The authors could be a bit more clear about what the potential next steps should be, what key open questions are for operationalising train-test development
- Laying out the key reproting components: While they give some indication of what could be reported, and the challenges that lie within that, they could be a bit more clear about what exactly should be reported - but also appreciate that this is not the focus of the paper

**Support:**

4

---

> ### Author Rebuttal · Authors · 2025-03-30
>
> Thank you so much for your kind and thoughtful review! We really appreciate your insight. To address the weaknesses:
>
> > The authors could be a bit more clear about what the potential next steps should be, what key open questions are for operationalising train-test development
>
> Thanks for the suggestion; we agree that this is worth fleshing out and added details on potential next steps. We've added a section on this: "Future Work: Standardization of Train-Test Overlap" (see response to Reviewer a1ak, as ICML does not allow uploading updated paper).
>
> To provide more context on the challenge of deciding on a concrete and actionable proposal, we actually began with a significantly different position: that model providers should report train-test overlap on a standardized protocol with a standard set of benchmarks using a standardized script, which improves the comparability of these numbers. Unfortunately, while we made some inroads with this (several model providers, including Writer, Meta, and Cohere ran this protocol), we were ultimately not successful in convincing enough model developers to adopt this for this to be meaningful. As a result, we changed our position to be a more tractable and credible demand. Our current position states that developers should report train-test overlap on the benchmarks they have reported themselves, and provide sufficient detail about the protocol they choose to run. We had more success with this position, which informed our choice to bring it to our community and call for others to join us in working with model developers to release these statistics.
>
> Our view is that our position and supporting work lays the groundwork for further standardization, as it is difficult to get adoption for a standard reporting protocol in an area where model developers are not doing reporting. With continued progress through our existing approach to encourage voluntary disclosure, we intend to work on standardization in reporting to make statistics more comparable.
>
> > Laying out the key reporting components
>
>   Thanks for the suggestion; we agree and have added details on standardization in the section above ("Future Work") and have provided concrete instantiation details of a protocol in our work on this standardization (thoroughly detailed, though given 5000 character limit, only a snippet is provided below).
>
> -----------
>
> \section{Protocol}
>
> As the training data for many language models is \private, we propose a protocol (see \autoref{fig:framework}) to coordinate between the model provider (first party) and the external entity (third party) to report \trainTestOverlap statistics.
> Through this protocol, relevant \trainTestOverlap statistics can be made \public while respecting the access controls placed on the underlying training data.
>
> \subsection{Notation}\label{sec:notation}
>
> The training data, denoted $D_{\text{train}}$, is a set of examples where each $x_{\text{train}} \in $ is a sequence of tokens. The test data consists of one or more test sets $D_{\text{test}}$, where $D_{\text{test}}$ is a set of examples where each instance $(x_{\text{test}}, Y_{\text{test}}) \in D_{\text{test}}$ consists of an input sequence of tokens $x_{\text{test}}$ and a reference set of token sequences $y_{\text{test}} \in Y_{\text{test}}$ representing the ground truth response. For instance, for a Q\&A test set, input would be the question whereas reference are the answer choices. We concatenate the reference set into a single sequence, so we will refer to the reference set as a single reference token sequence $y_{\text{test}}$ hereafter.
>
> \subsection{Steps}\label{sec:differential_overlap}
> We divide the above described process of computing \trainTestOverlap metrics between a first-party actor with access to the training set and a third-party actor in the following manner:
> 1. The third-party actor(s) release test data, $\textbf{compute-instance-overlap-statistics}$, $\textbf{aggregate-overlap-statistics}$, $\textbf{aggregate-metrics}$ publicly to instantiate the protocol.
> 2. The first-party actor uses $\textbf{compute-instance-overlap-statistics}$ on their $D_{\text{train}}$ and publicly available test data to output instance-level overlap statistics.
> 3. The third-party actor uses $\textbf{aggregate-overlap-statistics}$ on the instance-level overlap statistics to produce test-set-level statistics, which are published as meaningful overlap information between $D_{\text{train}}$ and test data.
> In terms of costs, we distinguish $\textbf{compute-instance-overlap-statistics}$ as the computationally expensive step relative to $\textbf{aggregate-overlap-statistics}$ as training sets are often several terabytes of data. This enables rapid iteration with different aggregation methods on the instance-level overlap stats.

---

### Official Review · Reviewer_a1ak · 2025-03-03

**Significance:** 4
**Argument Clarity:** 4
**Rating:** 4
**Confidence:** 5

**Questions:**

Please see my review of weaknesses. If my concerns are addressed, I am willing to raise my score.

**Discussion Potential:**

4

**Paper Summary:**

This paper investigates the transparency practices of reporting train/test overlap among 30 prominent language models. It reveals that a significant number of models do not disclose sufficient details about the overlap between their training data and evaluation test sets. By systematically scoring each model based on criteria such as whether the training data is publicly available or if explicit overlap statistics are provided, the paper highlights a transparency gap in current evaluation methods. Ultimately, the paper calls for language model developers to either release their training data or publish detailed overlap statistics to ensure more trustworthy evaluation outcomes.

**Position:**

Yes

**Position In Title:**

Yes

**Related Work:**

4

**Strengths And Weaknesses:**

**Strengths**
1. It provides a comprehensive analysis of train/test overlap reporting practices across 30 language models, clearly exposing the transparency gap in current evaluation methods.

2. It effectively argues for the critical role of transparent reporting in ensuring valid model evaluations, supported by empirical evidence and well-defined scoring criteria.

**Weaknesses**

1. originality of contribution

In a field where several studies (e.g., Golchin & Surdeanu, Shi et al., Oren et al.) have already addressed the train/test overlap issue, this paper needs to clarify whether it merely reaffirms the existence of the problem or if it offers novel insights that set it apart from previous work such as detailed case analyses or proposals for standardized transparency evaluations. As it stands, the paper appears to lack differentiating insights compared to the existing literature.

2. recent concrete examples of the consequences of not reporting train/test overlap

The authors argue that failing to report train/test overlap can undermine the validity of evaluation results. While many acknowledge the importance of addressing train/test overlap (or data contamination), there seems to be an equally significant number who remain unconvinced. This skepticism may stem from the difficulty in precisely quantifying overlap—as accurately measuring train/test overlap is challenging and its definition remains ambiguous—and from practical experiences where generalization appears to hold up, for instance, through RL-based post-training approaches. To robustly support the authors’ position, it would be beneficial to include a summary of concrete, recent examples (from the past year) where the absence of train/test overlap reporting has led to tangible issues. Although the paper cites the GPT-4 Codeforces coding problem as an example, that case dates back to 2023 and may not fully represent the current state of affairs.

**Support:**

2

---

> ### Author Rebuttal · Authors · 2025-03-30
>
> Thank you so much for the thoughtful review! To address the weaknesses:
>
> > originality of contribution
>
> Thanks for raising this; unlike Golchin & Surdeanu, Shi et al., Oren et al., we focus on white-box methods. Additionally, we put forth a concrete proposal for how we can improve train-test overlap, which has already helped improve transparency in the space.
>
> To expand, since 2023, there has been much work in alternative methods, and comparatively little work on train-test reporting, and we believe that the latter merits more attention. Accordingly, we engaged leading models developers and pushed for change, which resulted in publications of new train-test overlap statistics from three additional models. To our knowledge, this is the first such engagement from the academic community.
>
>   Our unique insight is derived from nearly two years of working with companies to make them more transparent on train test overlap.  This comment really helps us reflect about the original insights that we have gained from this process, and how to best communicate these learnings to the community.
>
>   To provide more context on the challenge of deciding on a concrete and actionable proposal, we actually began with a significantly different position: that model providers should report train-test overlap on a standardized protocol with a standard set of benchmarks using a standardized script, which improves the comparability of these numbers. Unfortunately, while we made some inroads with this (several model providers, including Writer, Meta, and Cohere ran this protocol), we were ultimately not successful in convincing enough model developers to adopt this for this to be meaningful. As a result, we changed our position to be a more tractable and credible demand. Our current position states that developers should report train-test overlap on the benchmarks they have reported themselves, and provide sufficient detail about the protocol they choose to run. We had more success with this position, which informed our choice to bring it to our community and call for others to join us in working with model developers to release these statistics.
>
>   We've added a section in the main paper (see below, as ICML doesn't allow uploading updated paper) which we supplemented thoroughly in the appendix to share some of the insights we gained through that experience, and to show the contrast that while communicating with companies and documenting train-test overlap seems simple, it can be highly effective in effectuating change. Hopefully, these experiences we share will help the community work together to better improve transparency in this space.
>
> We've also provided a snippet of the protocol in our response to Reviewer 2Bmm.
>
> > recent concrete examples of the consequences of not reporting train/test overlap
>
> Thanks for the suggestion; we have added a section on this with multiple recent examples of the consequences of not reporting train-test overlap. (see response to Reviewer GYw4)
>
> ---
>
> \section{Future Work: Standardization of Train-Test Overlap}
> \label{sec:future}
>
> The first critical step toward addressing train-test overlap is establishing a baseline expectation for model developers to publish overlap statistics for the benchmarks they themselves report, which we have advocated for here. We have seen promising adoption of this by nine model developers, including three new model developers from outreach. As this expectation becomes increasingly a reality, we have substantial additional work to further increase train-test overlap transparency.
>
> In particular, the model developers employ different methodology for computing train-test overlap statistics, which make it challenging to compare statistics between models. In fact, train-test overlap statistics can differ even between models for a given model developer, without clear reasoning. As such, we believe that standardization of the protocol to compute train-test overlap will be an essential next step to improving transparency and trust in the space.
>
> Here, we document an initial effort where we advocated for a standardized protocol requiring developers to compute overlap using shared benchmarks, tools, and scripts to ensure comparability (see Appendix for more details). While this approach garnered interest among a few model developers including Writer, Meta, and Cohere, widespread adoption proved challenging due to a lack of incentives to promote transparency.
>
> Our hope is that the community can learn from our efforts in standardization to better improve transparency in this space. Our view is that as community awareness of and discussion of train-test overlap becomes increasingly prevalent, model developers will have stronger incentives to release train-test overlap statistics and work toward standardization. With our efforts, we have taken initial steps in this direction, but there is still a significant amount of work to be done.

---

> > ### Comment · Reviewer_a1ak · 2025-04-05
> >
> > Thank you to the authors for their responses. My concerns have been resolved; therefore, I have raised my score. I hope that one day, transparency regarding the test set will become a standard practice in language model development.

---

### Official Review · Reviewer_GYw4 · 2025-03-13

**Significance:** 3
**Argument Clarity:** 3
**Rating:** 3
**Confidence:** 4

**Questions:**

- The last few sentences in "Meta" paragraph in page 4 seem ill-formated.
- In page 6 when referring to Table 2, the "Table" is missing.

**Discussion Potential:**

2

**Paper Summary:**

This paper advocates that LLM developers should report the train-test overlap statistics. They study 30 widely used LLMs on whether they report the train-test overlap statistics and found that only 9 models reveal enough information that allow users to understand the train-test overlap of the model. The paper also discuss the difficulty in evaluating the train-test overlapping and categorize some common train-test overlapping found in the Pile. The paper also provides some alternate method for estimating and addressing train-test overlapping, while all alternative methods have drawbacks and is less effective comparing to the train-test overlap reported by the LLM developers themselves.

**Position:**

Yes

**Position In Title:**

Yes

**Related Work:**

2

**Strengths And Weaknesses:**

Strengths
===
1. The paper is well-written. The position it stands is clear and the paper provides interesting evidence of how LLM developers report or do not report train-test overlap.
2. The papers author contacts the developer of widely used LLMs and prompt them to report or conduct analysis on train-test overlap. It's good to see that the authors' effort really encourage some model developers to report how they measure train-test overlap.


Weaknesses
===
1. To justify the necessity of reporting train-test overlap, I think the paper can allocate more efforts on "what will happen if model developers do not do this". I think this is only slightly discussed in the last paragraph in page 1, but this can use a full section to justify the need of doing this. I believe the paper has a lot of additional space for doing this.
2. The content of this paper is not very rich. For example, while it summizes the effort of prior papers that conduct train-test decomtaminitation, I find the contents in Section 4.3 that are referenced from other papers in verbatim somewhat uninformative. I believe those information is important, but I am not sure if this should be presented in the main content.
3. I don't think this paper can inspire constructive, useful discussion in the ICML community since the position stated in this paper should be well-accepted. I suspect anyone will object to the position of this paper. I mark this as a weakness since I am not sure what are the new things this paper can bring to the community.

**Support:**

2

---

> ### Author Rebuttal · Authors · 2025-03-30
>
> Thank you so much for the thoughtful review! We've made all the updates and suggestions you've made (new section on "what will happen if model developers do not do this", moved the contents in Section 4.3 to Appendix, and fixed the formatting/missing word). As ICML doesn't allow uploading updated paper, see the new section pasted below for reference. Also, we address the question about the position.
>
> > I suspect anyone will object to the position of this paper.
>
> Our position can be restated as "other approaches (hidden test set, black box detection) for train-test contamination are insufficient and that public disclosure by the model providers is critical for transparency in this space". Phrased this way, we do think that some people would take the opposing view, as the efforts have been focused on other approaches. Additionally, we put forth a concrete proposal for how model developers can improve this, which could be a point of discussion (i.e. other members of the community suggestions to improve the proposal / offer other proposals).
>
> To expand, since 2023, there has been much work in alternative methods, and comparatively little work on train-test reporting, and we believe that the latter merits more attention. Accordingly, we engaged leading models developers and pushed for change, which resulted in publications of new train-test overlap statistics from three additional models. To our knowledge, this is the first such engagement from the academic community.
>
> We also provide more details on the challenge of deciding on a concrete and actionable proposal in response to Reviewer a1ak (new section on Future Work: Standardization of Train-Test Overlap, outlining alternative approach we took).
>
> ---
>
> \section{Consequences of Not Reporting Train-Test Overlap}
> \label{sec:consequences}
>
> Failing to disclose \trainTestOverlap can degrade trust within both the AI community and the broader public, ultimately diminishing the value of evaluations that are pivotal for research progress and real-world applications. The current landscape of AI evaluation is increasingly characterized by accusations of "cheating" or undisclosed advantages, fostering a climate of low trust within the community . This lack of transparency not only undermines the credibility of individual model evaluations but also has broader negative consequences for the advancement and trustworthiness of AI as a whole.
>
> One significant consequence of not reporting train-test overlap is that it leads to unsubstantiated claims by model developers, and consequently an erosion of trust. For example, OpenAI initially reported that GPT-4 had achieved state-of-the-art performance on a test set of Codeforces coding questions, claiming that there was no contamination \citep{openai2023gpt4}.
> Yet it was later demonstrated\footnote{See \url{https://twitter.com/cHHillee/status/1635790330854526981?lang=en}} that while GPT-4 achieves 100\% accuracy for 10 pre-2021 problems, the model achieves 0\% accuracy on more recent problems. More recently, Anthropic claimed a "zero to one" moment in solving Capture The Flag tasks \citep{anthropic_strategic_warning_2025}. Yet Transluce found that Claude 3.5 solves broken tasks by memorizing answers and hallucinating them \citep{mengk20_tweet_2025}. Instead, if the model developers had provided train-test overlap statistics associated with these results, it would have been clear that train-test overlap was a significant contributor of these breakthrough results. Accordingly, it increases accountability of model developer, which then increases the trust in their claims.
>
> Perhaps even more significant than these instances with clear evidence are accusations of cheating that are hard to verify, which end up eroding trust further. For instance, there are "cheating" accusations against o3 on the ARC-AGI benchmark \citep{lesswrong_arc_agi_2024}, which OpenAI has denied. While we can attempt to derive evidence via black-box methods \citep{golchin2023time,shi2023detecting,oren2023proving}, they have limitations and cannot fully substitute for transparent reporting from model developers. The inherent uncertainty associated with black-box methods, coupled with the potential for adversarial manipulation, means that accusations and counter-accusations can persist, leading to a fragmented and less productive research environment.
>
> Accordingly, the failure to report train-test overlap has significant negative consequences for the AI community. It fosters a climate of distrust, hinders scientific progress by obscuring the true nature of model capabilities, and makes it difficult to build upon existing research. By embracing transparency and routinely reporting train-test overlap statistics, the AI community can move towards a more trustworthy and collaborative environment that ultimately accelerates the development of safer and more trustworthy AI.

---

### Official Review · Reviewer_AP1m · 2025-04-02

**Significance:** 3
**Argument Clarity:** 4
**Rating:** 4
**Confidence:** 3

**Questions:**

N/A

**Discussion Potential:**

4

**Paper Summary:**

The authors summarize existing methods to estimate and address train-test split, concluding that they are not sufficient to deal with this important issue in machine learning. They offer concrete information in existing models, and urge the community to publish train-test overlap.

**Position:**

Yes

**Position In Title:**

Yes

**Related Work:**

4

**Strengths And Weaknesses:**

The author tackle a revelant topic, and provide compelling arguments for their position. Table 1 contains very important information about several models and Table 2 illustrates some examples. While the suggestions/position are straightforward, they are nonetheless well supported.

Some minor typos (line 58) distract from the reading flow, but overall flow is good.

**Support:**

3

---

### Decision · Program_Chairs · 2025-04-30

**Decision:**

Accept (spotlight poster)

**Comment:**

There is consensus among the reviewers that this paper's viewpoint is important and relevant for the community.  The authors responded to reviewer questions / weaknesses with satisfactory clarifications and proposed additions to the paper.

I want to highlight a particular strength of the work, which is the engagement of the authors with model developers to assess transparency, giving them a chance to respond and correct any mistaken conclusions.

Authors should carefully check the references and replace arXiv citations with the peer-reviewed versions, when available (e.g., Kapoor & Narayanan, 2022 -> Kapoor & Narayanan, 2023 in Patterns).